# Organic Food Consumption among Households in Hanoi: Importance of Situational Factors

**Anh Thi Van Tran**  **and Nhung Thi Nguyen ***

Faculty of Finance and Banking, VNU University of Economics and Business, Hanoi, Vietnam;
ttvanh@vnu.edu.vn
* Correspondence: ntnhung@vnu.edu.vn

**Abstract:** The promotion of sustainable consumption plays an extremely important role in Vietnam's National Green Growth Strategy. However, despite an increase in concerns about environmental issues, eco-friendly buying behaviors in general and organic food consumption in particular are still unpopular among Vietnamese consumers, leading to a question about the importance of situational factors, which this article focuses on. Based on attitudes; subjective norms; perceived norms, which are mentioned in the theory of planned behavior (TPB) and consumer choice theory; and social norms from social categorization theory, the research created a questionnaire and then sent it to respondents who were in charge of buying food for their family in Hanoi. Then, 423 of the 570 responses received were used to create the structural equation model (SEM) with four distinct stages in AMOS statistical software, which evidences the crucial role of situational factors. Subjective norms and social norms have the highest positive impact on organic food purchase among households in Hanoi. Moreover, households' organic food purchase is also positively affected by perceived behavior control and the availability of products but negatively affected by the price of products, which strongly fits with TPB, social categorization theory, and other studies. In particular, there is no evidence about relationships between knowledge of or attitude toward organic food and family income and organic food purchase among households in Hanoi. In addition, age and education status do not have any impact on respondents' behavior in organic food purchase in this city. Finally, the authors propose some suggestions to promote organic food consumption among households in Hanoi. First is that businesses specializing in the production and supply of organic products should focus on customer care activities, innovate business models, and advertise to attract customers to use organic products. Second is that the government should issue regulations to encourage businesses to invest in the research and production of organic products as well as implement strict regulations to penalize violations in the production and supply of organic products.

**Keywords:** green consumption; organic food; green consumption gap; green lifestyle; consumer behavior theories

## 1. Introduction

The concept of green consumption may have emerged as early as the 1960s [1,2]. However, ooowHothe term "green consumption" has been used differently and sometimes interchangeably with other terms [1]. Green consumption is a form of consumption that allows people to participate in environmental protection [3] or is often referred to as 5R consumption, including reduce, revaluate, reuse, recycle, and rescue [4]. There are several terms related to green consumption, namely environmentally friendly products and environmentally responsible consumers. Environmentally friendly products are typically durable, non-toxic, made of recycled materials, or minimally packaged [5] or do not pollute the earth or damage natural resources and can be recycled or conserved [6]. Environmentally friendly products can, for our purposes, be divided into four categories, namely food and drink, domestic products, personal products, and green community [7,8].

According to Sun et al. [3], research on green consumption is focused on green consumption [9,10], green consumption patterns [11,12], green consumption marketing strategies, and the factors that influence green consumption [13,14].

The most frequently cited theory on the attitude/intention vs. behavior relation is the theory of planned behavior (TPB) [15]. This view is shared by Aertsens et al. [13] in their study providing an overview of studies on organic food consumption. Aertsens et al. [13] commented that previous studies on organic food consumption used the values theory of Rokeach [16] and Schwartz [17], but recent studies often use the theory of planned behavior. There are also other theories, which have been developed to explain the gap between consumer attitude/intention and behavior such as the theory of reasoned action (TRA) [18], norm-activation theory [19], and value–belief–norm (VBN) theory [20]. These models explain the attitude/intention and impact on the antecedents (i.e., value, attitude, beliefs, and norms) of the green consumer behavioral intention toward the actual green behavior. Some other researchers like [21–24] considered intention to be a factor in predicting behavior. An intention might be considered as people's willingness to perform certain actions [25], and a behavioral intention was the closest indicator to a real behavior. The stronger the intention to perform an action was, the higher was the probability that the behavior would be executed. In general, the terms attitude and intention are often used interchangeably.

In addition, these theories are also used as the basis for many studies that aim to explain the behavior of green customers. It is a situation that occurs when many customers have not altered their general behaviors, lifestyle, or purchasing decisions despite significant changes in the levels of awareness toward environmental and social problems. Even if attitudes toward green products result in the intention to alter an individual's purchasing behaviors, they do not necessarily translate into an actual change [26,27]. To be precise, Park and Lin noted that over 35% of survey respondents exhibited high positive purchase intention but failed to engage in purchasing these products [28]. Many consumers were aware of more environmentally friendly products when shopping, but few followed through and purchased the green products [29]. A majority of consumers (67%) had a positive attitude toward purchases of green products, but only a small proportion of them (4%) actually purchased those products [11]. Between 30% to 50% of people expressed their intention to buy green products; however, the market share of sustainable products was less than 5% of sales [30]. The percentage of people who declared that they were "green" was much higher than that the real "green" purchasers [31,32].

The same situation happens in Vietnam, where green products only account for a small proportion in households' total expenditure, which implies a large gap between attitude and behavior among Vietnamese households. In fact, promoting sustainable consumption as well as environmentally responsible lifestyles was an important task outlined in Vietnam's National Green Growth Strategy. A series of activities to promote green consumption were deployed from the level of state management agencies to enterprises. In the state management sector, the eco-labeling program of the Ministry of Natural Resources and Environment, the energy-saving program of the Ministry of Industry and Trade, the successfully organized annual campaign of green consumption in Ho Chi Minh City, and the Green Destination Network program held in Hanoi are all worthy of mention. From a business perspective, a typical example is the Ho Chi Minh City Trade Cooperatives Union (Saigon Co-op). This is the pioneering retailer to participate in the campaign "Green Consumption". Vietnamese consumers are increasingly concerned with environmental issues in general and concepts such as "green" and "clean" in particular. The general understanding of the causes of global warming, green products, and green consumption by household groups has been significantly improved in Vietnam, leading to an increase in eco-friendly buying behavior among Vietnamese consumers.

Our quick survey among Vietnam consumers about organic food as a trendy green product in Vietnam obviously showed that organic food was unpopular among households in Hanoi (Figure 1). To be precise, 32.15% of respondents said that they did not buy or

bought organic food infrequently, while these kinds of products were consumed by 67.75% of respondents' families in spite of differences in consumption frequency. In addition, 20.57% of respondents' families said that they consumed organic food every day or almost every day, followed by 14.89% households, who bought organic food 4–5 times a week. Families that bought organic food 2–3 times a week accounted for 22.70%, compared to 7.09% and 2.40% of respondents' families who used organic food once a week or several times a month.

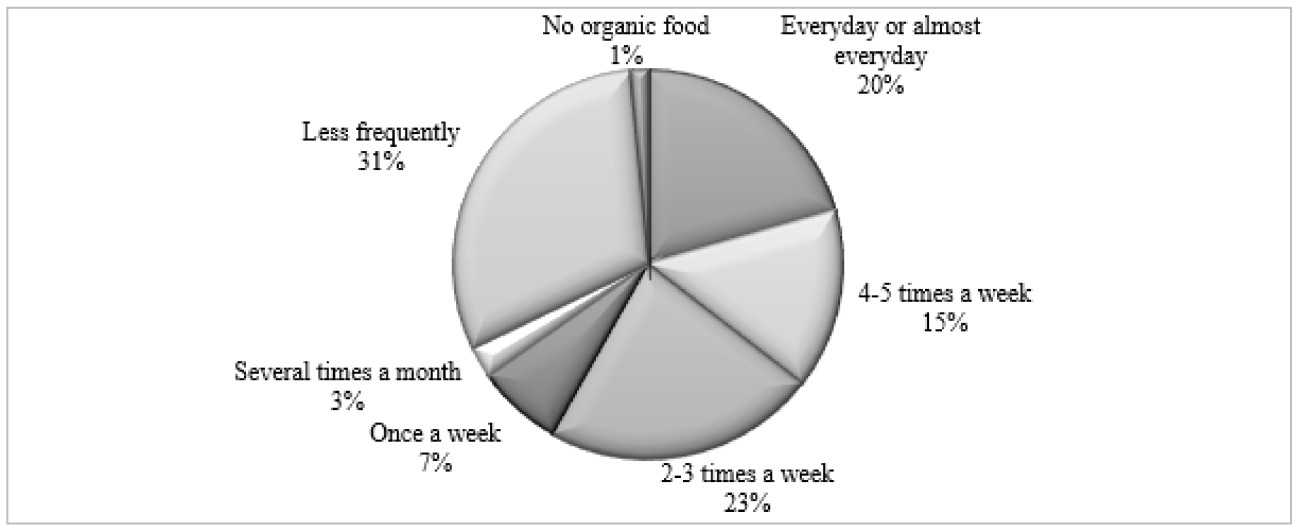

**Figure 1.** Frequency level of organic food consumption in households in Hanoi.

The above-mentioned survey results provided us with a high motivation to carry out this study. The purpose of our article is to determine factors influencing organic food purchase among households in Hanoi (Vietnam) and compare how much these factors impacted on this behavior in terms of the age and education level of respondents who were in charge of daily food purchase for their family. In particular, the research concentrates more on situational factors including households' income, price, and availability of organic food belonging to perceived behavior control as one of the three main components of the theory of planned behavior (TPB), as well as social norms derived from the social categorization theory, in order to give recommendations to promote environmentally responsible consumption.

To the best of our knowledge, this research is framed in contributing to the theoretical basis of consumer behavior and green consumption behavior as well as providing empirical evidence indicating determinants of organic food purchase in general and situational factors among households in Hanoi in particular. This is a topic that has not been mentioned by many research papers in Vietnam. For example, the paper of Nguyen et al. [33] applied the same research method as ours, but the variables used in Nguyen's model were not completely identical with the variables in our model. Therefore, the empirical results and the interpretations of Nguyen et al. were not exactly the same as ours. Therefore, our research results are still useful references in providing more efficient solutions for bridge this gap in the city.

After the introduction, which briefly presents the reasons for our research and its purposes, methodology and data used to achieve research objectives are presented in the second part. The third part explains relevant results, followed by the discussion and conclusion.



## 2. Methodology

### 2.1. Developing the Questionnaire

Development of the questionnaire was the first stage of determining the research model. The research mainly aimed to determine factors influencing organic food purchase among households in Hanoi (Vietnam) and compare how much these factors impacted on this behavior in terms of the age and education level of respondents who are in charge of daily food purchase for their family. In particular, the research focused much more on situational factors including households' income, price, and availability of organic food, which belongs to perceived behavior control—one of the three main components of the theory of planned behavior (TPB)—as well as social norms, which primarily originated from social categorization theory.

It can clearly be seen that the TPB is one of the most influential theories for the prediction of ecological behaviors, because it takes into consideration socio-cultural factors with three principal components, including attitude, subject norms, and perceived behavior control.

In theory, an "attitude" is the predisposition of the individual to evaluate a particular object in a favorable or unfavorable manner [34] or an enduring set of beliefs about an object that predisposes people to behave in particular ways toward the object [35] or the result of a consumer's assessment of particular behavior [15]. Moreover, attitudes toward a certain behavior are mostly influenced by knowledge since individuals can change their purchasing behaviors to buy more eco-friendly goods if they understand climate change or they are concerned about this issue. Consumers' knowledge is mentioned by many previous studies [22,24,36], so our study used knowledge as the main factor that had a direct impact on attitude toward organic food and an indirect effect on consumers' organic food purchase as a mediator of attitude.

Subjective norms raise the question of consumer's trust in the opinions of people around them and how this affects their purchasing behavior concerning the products they consume. This component is already mentioned in different previous studies [24,36].

Perceived behavior control is defined in this paper as the consumer's own assessment of how difficult or easy it is to purchase organic products. The research focused on three different main factors, including households' income, price of products [24,28,37,38], and availability of products [38–40]. In fact, income, price, and availability of products are primarily derived from the consumer choice theory, showing that people decide to buy a product based on their individual preferences and budget. In fact, consumers always consider these factors a lot before making choices. Moreover, availability of products requires a wide variety of products and their accessibility. However, the consumer theory is based on a full set of assumptions about human behavior. For instance, consumers can be irrational when deciding to buy a product that is expensive but environmentally friendly. In particular, this research also tested the indirect effects of three situational factors on organic food purchase among households in Hanoi through perceived behavior control.

Furthermore, it can be obviously seen that there are several limitations of the TPB since it still does not take into account environmental or economic factors that may influence a person's intention to perform a behavior. Consumers may also lack confidence that producers truly adhere to the principles of respect for the environment [41], while attitude and intention–behavior gap size depend on the tangibility of specific economic or symbolic nature benefits, which may increase the consumers' desire to choose to save energy and pursue an eco-lifestyle [42]. Therefore, this research referred to the social categorization theory, which concentrated on social norms for consumers' environmental behavior [28].

The research model is presented in Figure 2.

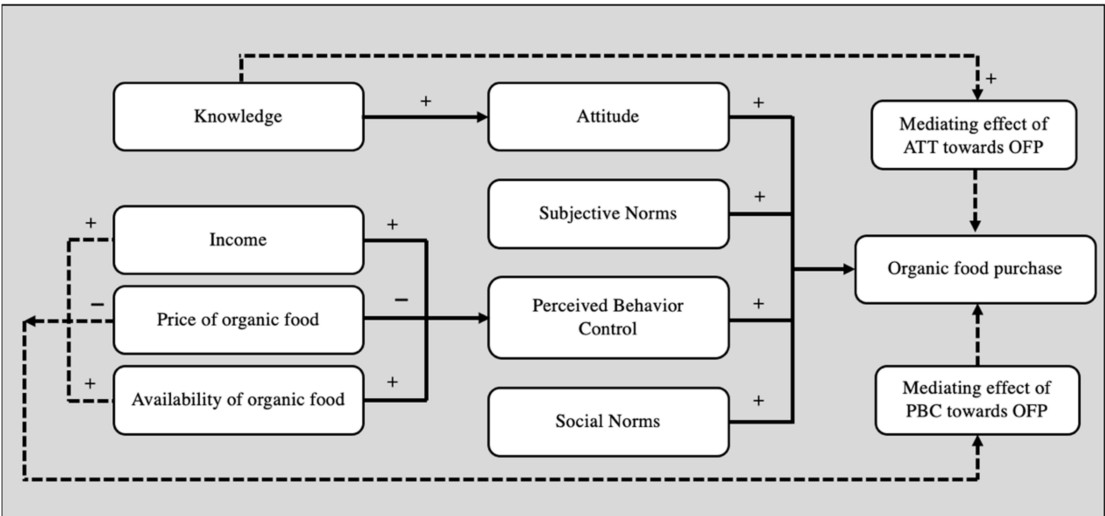

**Figure 2.** Conceptual framework.

In the next steps, the authors propose a number of questions based on the research model. An online survey included two main parts, namely respondent profile and survey content. The first one asked interviewees to give general information about their individual demographics such as education, income, or employment. In particular, there were two questions to verify whether respondents resided in Hanoi and were in charge of buying food for their family since this research focused on the households' consumption. The second part had four main questions. Due to the fact that respondents always overestimated their attitudes because of social desirability bias, the survey's first question was aimed at verifying respondents' knowledge of organic food. The second question explored the present state of behavior toward organic food consumption among households in Hanoi. The importance of factors leading to organic food consumption given by respondents was presented in the third question. The last question included a series of sub-questions focusing on respondents' attitude, subjective norms, perceived behavior control, and organic food's price and availability as well as social norms.

To ensure the accuracy of responses, the research used various kinds of questions including close-ended and open-ended questions as well as Likert-scale questions with a five-point scale from 1 to 5, which allowed the individual to express how much they agreed or disagreed with a particular statement (Supplementary Material Figure S1, "SURVEY CONTENT"). Moreover, to make sure that questions were comprehensible to the respondents as well as to ensure that there was no bias, this survey was initially sent to some housewives and organic food shop managers in Hanoi. Based on their feedback, all questions were carefully revised and shortened before posting on social media (Facebook and Gmail).

### 2.2. Data Collection and Methods of Data Analysis

Data collection was carried out from the beginning of May until the ending of July 2021 (two months). As the data collection phase was coming to an end, the researchers successfully received a total of 570 respondents but only accepted 423 responses, which was totally appropriate for structural equation modeling (SEM).

There were 12 hypotheses to be verified, as follows:

**Hypothesis 1 (H1).** *Knowledge (KNL) has a positive impact on attitude toward organic food purchase (ATT) among households in Hanoi.*

**Hypothesis 2 (H2).** *Price of organic food (PRI) has a negative impact on perceived behavior control toward organic food purchase (PBC) among households in Hanoi.*

**Hypothesis 3 (H3).** *Availability of organic food (AVA) has a positive impact on perceived behavior control toward organic food purchase (PBC) among households in Hanoi.*

**Hypothesis 4 (H4).** *Households' income per person (INC) has a positive impact on perceived behavior control toward organic food purchase (PBC) among households in Hanoi.*

**Hypothesis 5 (H5).** *Attitude toward organic food (ATT) has a positive impact on households' organic food purchase (OFP) in Hanoi.*

**Hypothesis 6 (H6).** *Subjective norms (SUN) have a positive impact on households' organic food purchase (OFP) in Hanoi.*

**Hypothesis 7 (H7).** *Perceived behavior control (PBC) has a positive impact on households' organic food purchase (OFP) in Hanoi.*

**Hypothesis 8 (H8).** *Social norms (SON) have a positive impact on households' organic food purchase (OFP) in Hanoi.*

**Hypothesis 9 (H9).** *Attitude toward organic (ATT) food mediates the effect of knowledge (KNL) on households' organic food purchase (OFP) in Hanoi.*

**Hypothesis 10 (H10).** *Perceived behavior control (PBC) mediates the effect of price of organic food (PRI) on households' organic food purchase (OFP) in Hanoi.*

**Hypothesis 11 (H11).** *Perceived behavior control (PBC) mediates the effect of availability of organic food (AVA) on households' organic food purchase (OFP) in Hanoi.*

**Hypothesis 12 (H12).** *Perceived behavior control (PBC) mediates the effect of households' income per person (INC) on households' organic food purchase (OFP) in Hanoi.*

To complete the research objective, we used structural equation modeling (SEM) in AMOS with four distinct stages.

First of all, data were examined by Cronbach' alpha in SPSS. Cronbach' alpha is considered to be a measure of scale reliability. A reliability coefficient of 0.70 is considered "acceptable". Simultaneously, data must have corrected item–Total correlation equal to or larger than 0.3. In particular, in case the previous condition is satisfied, but the Cronbach alpha of Item Deleted is greater than Cronbach' alpha, data should be verified carefully. After testing Cronbach's alpha in SPSS, the research only kept appropriate factors by removing unsuitable variables from the data. Suitable variables were introduced in SPSS to test EFA.

Secondly, the exploratory factor analysis computed variables in appropriate groups. Correlation is a bivariate analysis that measures the strength of association between two variables and the direction of the relationship. In terms of the strength of relationship, the value of the correlation coefficient varies between $+1$ and $-1$. In this research, the authors used Pearson correlation to measure the degree of the relationship between linearly related variables.

Thirdly, the research used the confirmatory factor analysis to test whether measures of a construct were consistent with the research's understanding of the nature of factors or whether the data fit a hypothesized measurement model.

Finally, the structural equation modeling (SEM) was used to analyze structural relationships between measured variables and latent constructs. The reason for choosing SEM was that this model allows to estimate of dependence in a single analysis, which is highly appropriate for behavioral analysis. In addition, SEM was used to compare research results for respondents from three different age groups (21–34 years, 35–49 years, and others)

as well as three groups with distinct education levels (postgraduate, bachelor's degree, and others).

## 3. Research Results

### 3.1. Descriptive Statistics

Table 1 shows statistics about the respondents, of whom 333 (78.72%) were women and the remaining were men. Moreover, 46.34% of the respondents were from Generation X (35–49 years old), followed by 30.26% of millennials (21–34 years old). Generation Z, in the age from 15 to 20 years, accounted for 13.95%, while the remaining 8.45% were baby boomers (50–64 years old) and the silent generation (over 64). A majority of the respondents had completed bachelor's degree (50.12%) and postgraduation (33.10%), while 16.78% of them had completed high school or less.

**Table 1.** Demographic profile of respondents.

| Gender | | | Education Status | | |
|---|---|---|---|---|---|
| | Count | Percent | | Count | Percent |
| Men | 90 | 21.28% | High school or less | 71 | 16.78% |
| Woman | 333 | 78.72% | Bachelor's degree | 212 | 50.12% |
| Total | 423 | 100.00% | Postgraduate | 140 | 33.10% |
| | | | Total | 423 | 100.00% |
| **Age** | | | **Occupation** | | |
| | Count | Percent | | Count | Percent |
| 15–20 | 59 | 13.95% | Student | 80 | 18.91% |
| 21–34 | 128 | 30.26% | Employed | 278 | 65.72% |
| 35–49 | 196 | 46.34% | Self-employed | 41 | 9.69% |
| 50–64 | 27 | 6.38% | Unemployed | 3 | 0.71% |
| Over 64 | 13 | 3.07% | Retired | 21 | 4.96% |
| Total | 423 | 100.00% | Total | 423 | 100.00% |
| **Marital status** | | | **Household Income** | | |
| | Count | Percent | | Count | Percent |
| Single or divorced, live alone | 64 | 15.13% | Less than VND 10 million | 67 | 15.84% |
| Single or divorced, live with parents, children, siblings, or friends | 88 | 20.80% | From VND 10 million to 20 million | 102 | 24.11% |
| Married without children | 9 | 2.13% | From VND 20 million to 40 million | 137 | 32.39% |
| Married with children | 262 | 61.94% | From VND 40 million to 60 million | 50 | 11.82% |
| Total | 423 | 100.00% | More than VND 60 million | 67 | 15.84% |
| | | | Total | 423 | 100.00% |

USD is equivalent to about 22,750 VND.

For occupation status, 65.72% of the respondents were employed, while 9.69% were involved in entrepreneurial activity. Students and retired people accounted for 18.91% and 4.96%, respectively, and 0.71% were unemployed.

In terms of respondents' marital status, most of them were married and lived with their children (61.94%), followed by 20.80% people who were single or divorced and lived with parents, children, siblings or friends. In all, 15.13% of respondents were single or divorced but lived alone, compared to only 2.13% of people who got married but had not yet had children.

Regarding household income, 32.39% families disposed of between VND 20 million and 40 million (1 USD was equivalent to about 22,750 VND) per month, followed by 24.11% families whose monthly income ranged from VND 10 million to 20 million. Furthermore, 15.84% of respondents' families had monthly earnings of more than VND 60 million or less than VND 10 million, while only 11.82% of families earned from VND 40 million to 60 million.

### 3.2. Empirical Results

3.2.1. Assessment before Structural Equation Modeling (SEM)

Cronbach' alpha, which was applied in SPSS for factors related to knowledge, attitude toward organic food purchase, subjective norms, perceived behavior control toward organic food purchase, price of organic food, availability of organic food, households' income per person, and social norms, indicated that all factors are reliable and appropriate to test exploratory factor analysis. Results of Kaiser–Meyer–Olkin (KMO) test of 0.805 and Bartlett's test of sphericity of 0.000 meant that it was possible to proceed with a satisfactory factor analysis. In addition, an extraction sums of squared loadings (cumulative%) of 62.327% (bigger than 50%) indicates the appropriateness of exploratory factor analysis model. The significance of all variables was bigger than 0.05 meaning that there was no linear relationship between variables (Supplementary Material Table S1–S10).

In the next stage, confirmatory factor analysis was executed to investigate the reliability and validity of the measurement instruments. As a result, minimum discrepancy per degree of freedom (CMIN/DF) was 1.041, which is smaller than 3. Goodness-of-fit index (GFI), comparative fit index (CFI), and Tucker–Lewis index (TLI) were 0.970, 0.998, and 0.998, respectively, which are all bigger than 0.9. Moreover, the root mean squared error approximation (RMSEA) of 0.010 was bigger than 0.08, while the *p* value for testing the null hypothesis of close fit (PCLOSE) of 1.000 was bigger than 0.05. This means that all indicators of model fit were totally appropriate to the research model. Furthermore, all standardized regression weights bigger than 0.5 were meaningful in confirmatory factor analysis [43].

Furthermore, the factors' composite reliability and average variance extracted value were all greater than 0.7 and 0.5, respectively, which leads to an appropriate convergent validity of the scale [43]. Moreover, the square roots of average variance extracted values were greater than the corresponding factors' correlations between latent variables, and the maximum shared squared variance was smaller than the average variance extracted value. This indicated discriminant validity [44].

3.2.2. Direct and Indirect Effects

Figure 3 shows results of structural equation modeling (SEM). In terms of regression weights, *p*-values of relationships between households' income per person and perceived behavior control toward organic food purchase, and attitude toward organic food purchase and households' organic food purchase were 0.289 and 0.100, which are bigger than 0.05, meaning that households' income per person and attitude toward organic food purchase did not have any impact on perceived behavior control toward organic food purchase and households' organic food purchase, respectively. In other words, hypotheses 4 and 5 were rejected and the remaining hypotheses were supported.

Table 2 gives information about regression weights, standardized regression weights, and squared multiple correlations. In terms of standardized regression weights, the bigger the estimate value was, the more significant wa the impact that independent variables had on dependent variables. Therefore, it can clearly be seen that knowledge had the most positive influence on attitude toward organic food purchase, followed by subjective norms, social norms and perceived behavior control, which impacted households' organic food purchase. In particular, availability of organic food had a positive effect on perceived behavior control (with an estimate of 0.234), while the price of organic food had an opposite impact on perceived behavior control (with an estimate of −0.241). Regarding squared multiple correlations, perceived behavior control, attitude toward organic food purchase, and households' organic food purchase had an R2 value of 0.116, 0.342, and 0.589, respectively. This meant that the independent variables perceived behavior control, attitude toward organic food purchase, and households' organic food purchase could explain 11.6%, 34.2%, and 58.9% of change in perceived behavior control, attitude toward organic food purchase, and households' organic food purchase, accordingly.

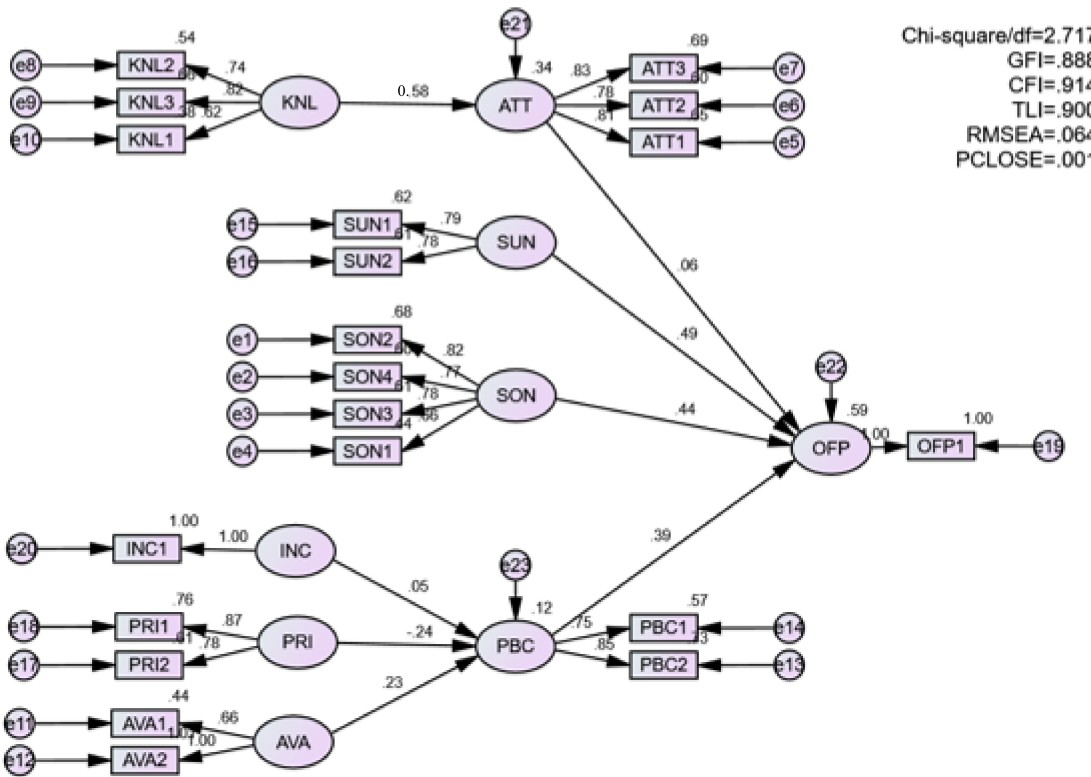

**Figure 3.** Results of SEM about determinants of households' organic food purchase in Hanoi.

**Table 2.** Regression weights, standardized regression weights, and squared multiple correlations.

| Regression Weights | | | | | | | Standardized Regression Weights | | | |
|---|---|---|---|---|---|---|---|---|---|---|
| | | | Estimate | S.E. | C.R. | p | | | | Estimate |
| ATT | ← | KNL | 0.594 | 0.065 | 9.188 | | ATT | ← | KNL | 0.585 |
| PBC | ← | INC | 0.034 | 0.032 | 1.061 | 0.289 | PBC | ← | INC | 0.055 |
| PBC | ← | PRI | −0.194 | 0.054 | −3.608 | | PBC | ← | PRI | −0.241 |
| PBC | ← | AVA | 0.273 | 0.062 | 4.370 | | PBC | ← | AVA | 0.234 |
| OFP | ← | SON | 0.380 | 0.036 | 10.568 | | OFP | ← | SON | 0.435 |
| OFP | ← | SUN | 0.412 | 0.041 | 9.977 | | OFP | ← | SUN | 0.490 |
| OFP | ← | ATT | 0.055 | 0.033 | 1.646 | 0.100 | OFP | ← | ATT | 0.063 |
| OFP | ← | PBC | 0.356 | 0.042 | 8.562 | | OFP | ← | PBC | 0.394 |

| Squared Multiple Correlations | |
|---|---|
| | Estimate |
| PBC | 0.116 |
| ATT | 0.342 |
| OFP | 0.589 |

Note: ATT: attitude toward organic food; PBC: perceived behavior control; OFP: households' organic food purchase; KNL: knowledge; INC: households' income per person; PRI: price of organic food; AVA: availability of organic food; SON: social norms; SUN:subjective norms.

Regarding mediating factors, Table 3 shows that (i) there was no significant effect of knowledge and households' income per person on households' organic food purchase through attitude toward organic food and perceived behavior control, respectively, since the *p*-values of 0.139 and 0.274 are bigger than 0.05; (ii) there was a negative indirect effect of price of organic food on households' organic food purchase through perceived behavior control (*p*-value of 0.001 and standardized estimate of −0.095); (iii) there was a positive indirect effect of availability of organic food on households' organic food purchase through perceived behavior control (*p*-value of 0.001 and standardized estimate of 0.092). In other words, hypotheses 9 and 12 were rejected.

**Table 3.** Mediating factors.

| Specific Indirect Path | *p*-Value | Standardized Estimate |
|---|---|---|
| KNL→ATT→OFP | 0.139 | 0.037 |
| INC→PBC→OFP | 0.274 | 0.022 |
| PRI→PBC→OFP | 0.001 | −0.095 |
| AVA→PBC→OFP | 0.001 | 0.092 |

A summary of the results can be found in Table 4. It can clearly be seen that hypotheses 4, 5, 9, and 12 were rejected meaning that households' income per person and attitude toward organic food did not have any impact on perceived behavior control and households' organic food purchase, respectively. In addition, knowledge and households' income per person did not have any effect on households' organic food purchase through attitude toward organic food and perceived behavior control, respectively.

**Table 4.** Summary of SEM results.

| Hypothesis | Path | Estimated Coefficient | *p*-Value | Hypothesis Test |
|---|---|---|---|---|
| H1 | KNL→ATT | 0.585 | | Supported |
| H2 | PRI→PBC | −0.241 | | Supported |
| H3 | AVA→PBC | 0.234 | | Supported |
| H4 | INC→PBC | 0.034 | 0.289 > 0.05 | Rejected |
| H5 | ATT→OFP | 0.055 | 0.100 > 0.05 | Rejected |
| H6 | SUN→OFP | 0.490 | | Supported |
| H7 | PBC→OFP | 0.394 | | Supported |
| H8 | SON→OFP | 0.435 | | Supported |
| H9 | KNL→ATT→OFP | 0.037 | 0.139 > 0.05 | Rejected |
| H10 | PRI→PBC→OFP | −0.095 | 0.001 | Supported |
| H11 | AVA→PBC→OFP | 0.092 | 0.001 | Supported |
| H12 | INC→PBC→OFP | 0.022 | 0.274 > 0.05 | Rejected |

Differences in how much these factors impact on organic food purchase in terms of the age and education level of respondents. Note: ATT: attitude toward organic food; PBC: perceived behavior control; OFP: households' organic food purchase; KNL: knowledge; INC: households' income per person; PRI: price of organic food; AVA: availability of organic food; SON: social norms; SUN: subjective norms.

By contrast, there was evidence of relationships between other factors. To be precise, knowledge had a positive impact on attitude toward organic food. Availability of organic food had a positive impact on perceived behavior control. Price of organic food had a negative impact on perceived behavior control among households in Hanoi. In terms of households' organic food purchase, subjective norms had the most significant positive effect on this behavior with an estimated coefficient of 0.490, followed by social norms (with an estimated coefficient of 0.435) and perceived behavior control (with an estimated coefficient of 0.394). Price of organic food and availability of organic food had opposite impacts on households' organic food purchase through herceived behavior control among households in Hanoi, with estimated coefficients of −0.095 and 0.092, respectively.

Table 5 shows SEM results when respondents are divided into different groups of education status (high school or less, bachelor's degree and postgraduate) and age (21–34, 35–49 and others). It can be clearly seen that *p*-value is equal to 0.526 and 0.656, bigger than 0.05. This means that there is no difference in Chi-square or how much these factors impact on organic food purchase in terms of the age and education level of respondents (Supplementary Material Figure S3–S6).

**Table 5.** SEM results according to respondents' education status and age.

| | Education Status | | | Age | |
| --- | --- | --- | --- | --- | --- |
| | **Chi-Square** | **Df** | | **Chi-Square** | **Df** |
| Immutable | 813.446 | 508 | Immutable | 822.329 | 508 |
| Mutable | 798.460 | 492 | Mutable | 809.105 | 492 |
| Difference | 14.986 | 16 | Difference | 13.224 | 16 |
| *p*-value | 0.526 | | *p*-value | 0.656 | |

## 4. Discussion

Subjective norms had the highest positive impact on organic food purchase among households in Hanoi, followed by social norms and perceived behavior control, respectively. In terms of subjective norms and perceived behavior control, the research results totally supported the theory of planned behavior (TPB) and were consistent with findings in studies such as [22–24,45]. In fact, Vietnamese consumers were mostly affected by people around them such as friends and family members. Moreover, situational factors played an extremely significant role in Vietnamese consumers' decision-making. This research strongly emphasized the importance of price and availability of organic food. To be precise, the price of such products had negative impacts on the purchase of organic food, while the availability of products had positive impacts on organic food purchase among households in Hanoi. According to our survey results shown in Figure 4 on consumers' evaluation about price and availability of food, approximately 70% respondents said that organic food was expensive in comparison with other products of the same kind, while 50% of them agreed that there was a variety in the range of organic food available. Only a minority of respondents mentioned that they could easily find organic food that they wanted to consume in trustworthy food retail outlets.

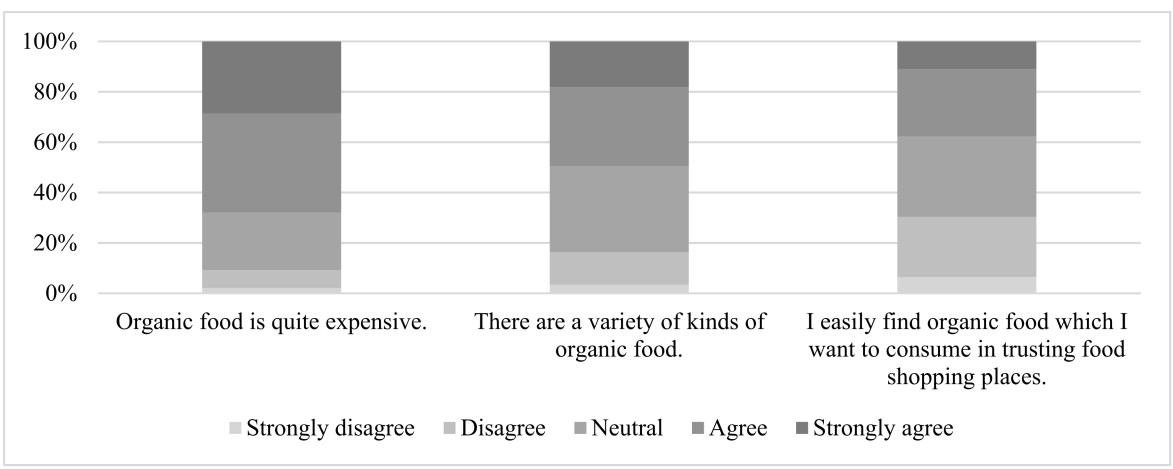

**Figure 4.** Consumers' evaluation about price and availability of food.

In Vietnam the price of organic food is on average 20–50% higher than the price of its conventional counterparts [46]. There are many reasons for this situation, for example, the number of businesses involved in organic food production is relatively small, so the supply of organic food is still limited compared to the demand. In addition, organic food production enterprises are often small scale, so labor costs and post-harvest processing and transportation costs are also higher than those for conventional foods [47]. Moreover, although the number of stores selling organic food has increased recently, it is still insufficient and not as widespread as conventional stores. To be precise, there are only 444 organic product chains in Vietnam that have been certified by the Ministry of Agriculture and Rural Development of Vietnam [47]. This shows that organic products are not as diverse or available when compared to conventional food products.

In addition, according to the results of our survey on consumers' evaluation about social norms related to organic food shown in Figure 5 social norms had a significantly positive effect on organic food purchase among households in Hanoi. Most respondents thought that organic food consumption is strongly supported by the Government, and there are regulations or law that punish people who sell unhealthy food. However, the current legal framework does not really encourage or support producers to deliver organic food, in addition to which, regulations related to organic food are not very efficient.

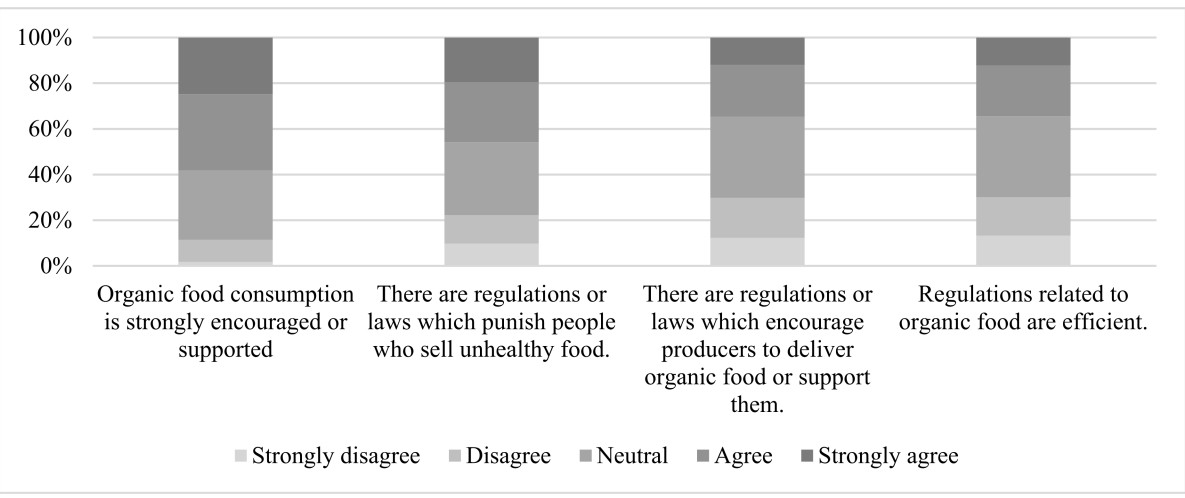

**Figure 5.** Consumers' evaluation about social norms related to organic food.

The National Assembly of Vietnam has promulgated a series of regulations forming a legal framework to support organic food consumption, such as the Law on Technical Standards and Regulations in 2006, Law on Product and Goods Quality in 2007, Law on food safety in 2010, or Law on Pharmacy in 2015. In particular, the Vietnamese Government also issued Decree No. 109/2018/ND-CP in 2018 on organic agriculture, which regulates the production, certification, labeling, logos, traceability, trading, and inspection of organic agricultural products in the fields of cultivation, animal husbandry, forestry, and aquaculture. Although these regulations have shown the Government's orientation toward green consumption, they have not yet achieved the desired effect since there is a lack of policies to strongly encourage the development of organic agricultural production.

Knowledge had a positive impact on attitude toward organic food. This result is totally in line with findings shown by [22,27,36]. In theory, individuals with high knowledge or awareness regarding environmental issues in general as well as concepts regarding "green" and "clean" in particular show a strongly positive impact on attitude toward organic food consumption. However, there is no evidence of relationships between knowledge or attitude toward organic food, as well as family income and organic food purchase among households in Hanoi, which are not totally consistent with the theory of planned behavior that says attitude toward green products plays an extremely crucial role in the intentions and behaviors of consumers. These results also do not support findings showed by [22–24,45]. In terms of the age and education status of respondents, there was no difference in how much these factors had an impact on organic food purchase. This means that age and education status did not have any impact on respondents' behavior in organic food purchase.

It can be obviously seen that research results related to the impacts of knowledge, attitude, education, and age of consumers on their behaviors in organic food purchase are inconsistent with theories related to consumption behavior and previous findings in other countries. To be precise, there was no relationship between what consumers understood about organic food or their attitude toward organic food and their behaviors in organic food purchase among households in Hanoi. They understood very well that organic food is clean, fresh (no preservatives), healthy, and close to nature as well as that organic

food consumption has positive impacts on human health and environmental protection. However, their behavior in organic food purchase was not correlated to their knowledge and attitude. This situation can be explained by the importance of situational factors mentioned above.

In particular, it is noticeable that the research results do not support respondents' perspectives about importance of factors, which are mentioned in Supplementary Material Figure S7. They believed that attitude toward organic food and households' income were the most significant factors (4.21 and 3.52) that impact organic food purchase, while subjective norms and social norms were considered less important factors (3.29 and 3.13). This would seem to indicate that people felt pressured to give an answer that they considered socially acceptable [48].

## 5. Conclusions

The research shows that organic food is not very popular among households in Hanoi. Moreover, this working paper emphasizes the importance of situational factors such as social norms as well as price and availability of organic food. This is the first time that situational factors have been evaluated and justified as key factors influencing Vietnamese customer behavior in organic food purchase. The empirical results are also the basis for the authors to propose recommendations to a number of stakeholders in order to promote organic food consumption among households in Vietnam in general and in Hanoi in particular.

Firstly, subjective norms are significantly important. Around 60% of respondents said that they trust food retail outlets because food origin is transparent, and these places are recommended by friends or family members that they trust. Therefore, enterprises specializing in the production and supply of organic products should pay special attention to developing customer care measures to establish a loyal customer network. To achieve this objective, businesses should concentrate on three key factors, namely product quality, consulting service quality, and after-sales service quality. Focusing on improving the service attitude of employees, regularly interacting with customers, and conducting surveys to record customer opinions and solve their problems in a timely manner are essential for businesses to increase reputation with customers. In addition, businesses should also launch promotions, especially for new products, to encourage customers to use these products.

Secondly, social norms are also important. Recently, the Vietnamese government has issued many legal documents to encourage green consumption. However, in order to increase the effectiveness of these policies, the government should introduce more specific regulations to encourage businesses to invest in the research and production of organic products, such as preferential policies on credit and land use tax. In addition, legal documents should have strict regulations dealing with violations in the production and supply of organic products.

Third is the importance of perceived behavior control. In the recent period, the Vietnamese consumer's awareness of organic food has increased. Especially in the context of the COVID-19 epidemic, the safety of organic products for health and the ability of these products to support the immune system, especially for the elderly and children [49], are some of the reasons that many customers prefer to use these products [46]. Therefore, in the future, businesses should conduct advertising programs to aim at raising the awareness of customers that the cost of using organic food is much less than the cost of treating the disease.

Fourth is the issue of the price of organic goods. A majority of respondents said that they were willing to pay extra for organic food. According to Kantar, 79% of Vietnamese people agree they are ready to pay a higher price for healthier foods [49]. However, when consumers pay higher prices for organic products, they want to find evidence and confidence to justify paying more. Therefore, it is important for businesses in the future to provide necessary information and factors to identify and distinguish organic products

from conventional products, e.g., product certifications and labels as well as product promotion materials. In order to increase customer confidence in products, they should also promulgate regulations for businesses on organic product information disclosure requirements as well as corporate responsibility for the accuracy of product information.

Regarding the availability of products, enterprises should implement an online and offline business model. The online channel may help to clarify issues regarding organic products for customers; thereby, allowing potential customers to find all the information needed to lead them to the store to make a purchase. Meanwhile, the offline channel will be the place to provide the most complete and satisfying experience for customers.

About 70% of the Vietnamese population (i.e., 68 million people) uses the Internet [50], leading to more and more Vietnamese consumers having the habit of using the Internet to search for information about products and services before making a purchase decision. In addition, nowadays, most people use a smartphone, so it is much more convenient to make online purchases or call and make an order directly. These are favorable conditions for businesses specializing in the production and supply of organic foods to develop online and offline business models. However, businesses themselves should also constantly strive to improve the quality of their operations to attract more customers in this competitive market.

Our study provided empirical evidence on the situational factors influencing consumer behavior in buying organic food in Hanoi, but the study still had some limitations. The research only collected 423 responses, which was unlikely to be representative of the full current situation of organic food consumption among households in Hanoi. Therefore, there is a need for further follow-up studies with a richer sample of data. Moreover, the questionnaire should be expanded with different questions about each factor, allowing researchers to better assess the degree to which factors have an impact on households' behaviors in organic food purchase in Hanoi, as well as provide highly feasible specific recommendations to thoroughly promote organic food consumption in Hanoi and in Vietnam.

**Supplementary Materials:** The following are available online at https://www.mdpi.com/article/10.3390/su132212496/s1, Figure S1: Respondents' profile., Figure S2: Results of CFA., Figure S3: Invariance., Figure S4: Mutability., Figure S5: Invariance., Figure S6: Mutability., Figure S7: Opinions of respondents about the importance of factors., Table S1: KNL., Table S2: PRI., Table S3: AVA., Table S4: ATT., Table S5: SUN., Table S6: PBC., Table S7: SON., Table S8: KMO and Bartlett's test., Table S9: Total variance explained., Table S10: Pattern matrix., Table S11: CMIN., Table S12: RMR, GFI., Table S13: Baseline comparisons., Table S14: RMSEA., Table S15: Standardized regression weights: (Group—Default model)., Table S16: Standardized regression weights.

**Author Contributions:** Conceptualization, A.T.V.T.; N.T.N.; methodology, A.T.V.T.; N.T.N.; software, N.T.N.; validation, A.T.V.T.; formal analysis, A.T.V.T.; investigation, A.T.V.T.; N.T.N.; resources, N.T.N.; data curation, N.T.N.; writing—original draft preparation, N.T.N.; writing—review and editing, A.T.V.T.; visualization, N.T.N.; supervision, A.T.V.T.; project administration, N.T.N.; funding acquisition, N.T.N. All authors have read and agreed to the published version of the manuscript.

**Funding:** This research was funded by University of Economics and Business (UEB)-Vietnam National University, Hanoi (VNU), grant number KT.19.05. The APC was funded by KT.19.05.

**Institutional Review Board Statement:** Not applicable.

**Data Availability Statement:** The data presented in this study are available on request from the corresponding author.

**Conflicts of Interest:** The authors declare no conflict of interest.

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
