# Peer review of "Organic Food Consumption among Households in Hanoi: Importance of Situational Factors"

_sustainability, doi:10.3390/su132212496_

Round 1

Reviewer 1 Report

Dear colleagues,

This is a very interesting contribution regarding the reasons leading to organic food purchase in a large urban city, Hanoi (Vietnam).

Although you have an adequate sample (more than 400 householders), your methodological approach is somewhat biased, due to several reasons:

  1. All these people are digitally fluent, and I suppose many other Hanoi inhabitants are not;
  2. There is also the possible positive relationship between your pro-active participants and relatively higher income, and I suppose these respondents are not among the most disfavored citizens of Hanoi;
  3. Your participants have voluntarily answered your invitation, and this means that the Organic Food subject was not indifferent to them, therefore it is possible that they are not representative of the population.

Besides, you have studied ‘representations of behaviors’, not ‘true behaviors’, and this is important in the Discussion and scope of the conclusions.

The manuscript is still quite informative; however, your results should be discussed bearing these factors in mind.

I will now state some of my main concerns in each section. Please check also the additional information included in the pdf.

ABSTRACT: Your abstract is not well-balanced. Please include one opening sentence stating the main interest of your work. The importance of the theoretical framework should be emphasized, clearly stating you are testing some hypotheses related with TPB, CCT and SCT and organic food consumption. The sentence where you address these theories needs an English revision, because the reader does not understand which of those focus “much more” on situational factors. Please reorganize information of the results so that the reader learns which factors affect and which factors do not affect the studied behavior.

INTRODUCTION: Your introduction is didactic, introducing several key-terms; however, a wider literature revision would be important to establish the importance of this theme, namely using the sustainability targets proposed by the United Nations and other international documents. Besides, you mention the research’ references, but you do not explain what the authors found, on which countries, target population, etc. (when appropriate), and why are their findings relevant to your research. Also, the third paragraph of the second page needs some more discussion. In particular, there are more ways a person may feel / be “green” apart from buying organic food.

METHODOLOGY: Apart from the first sentence, the first paragraph could be used in the introduction. Perhaps the second paragraph could start by stating which definition you are using in this study.

Social media – gmail and facebook are quite different media. How many people responded for each of the approaches? Are there any differences in the profile of the participants? Whose contacts were you using in the gmail? How many people did you contact / what was the response rate?

All the 12 hypotheses should be stated in Figure 1 – including (+) or (-) signs to signal the relationships you are testing.

When you mention Cronbach’Alpha, which scale are you referring to? How many items? Please expand this section. The explanation you give regarding the SEM is quite extensive, but you need to add some references.

RESULTS & DISCUSSION

Re-writing some of these paragraphs would greatly improve the clarity of the arguments. The hypothesis are quite important here, and results should be grouped considering the theories you are testing.

CONCLUSION

Your first conclusion affirms that organic food is not popular among Hanoi’ families. However, roughly two out of three respondents state that they consume organic food at least once a week. Is this so bad? Where are the studies that state that in other cities this consumption is higher?

You have used the references in “trains”, without explaining why you are using them, why they are relevant or which circumstances do they refer to. Please add some more details, so the reader may see the common ground among the situations.

FORMAL ISSUES

Language – the paper would benefit from a thorough English revision by a native speaker. Although the main ideas are understandable, these ideas are not presented in an elegant form, and the text it is quite repetitive on parts.

Legends of figures and tables – These are generally very poor and could be applied to any other study. Please state clearly what the models aim to represent, the number of participants and the date of the collection of data. The number of degrees of freedom stated in Table 5 for “immutable” is 508, which is higher than the 432 you stated were used for part of the analysis. How do you explain this? Please re-do the calculus or explain your methodology in a more comprehensive way.

Best wishes

Author Response

Dear the reviewer, 

I would like to thank you a lot for your valuable comments which we could learn a lot from. And please received our feedback for your comments.

Thank you a lot.

Authors

Reviewer 2 Report

The manuscript is expecting the factors influencing households’ behavior in organic food purchases in Hanoi, Vietnam.  TPB and SEM were chosen and employed to analyze 423 responses.  Conclusions drawn by the authors are reasonable and scientific.  The authors found subjective norms, social norms, and green marketing was important and necessary.  To my surprise, age and education status don’t have any impact on respondents’ behavior in organic food purchases in Hanoi.  Overall, the manuscript would merit publication in Sustainability, although some issues and questions listed below may have to be addressed first.  

  1. Why was the study carried out in Hanoi? What's the difference between this city among other cities in Vietnam?
  2. The concept of 5R consumption should be confirmed. Repurpose or revaluate?
  3. A questionnaire question for the organic food purchase ratio within the whole food purchase of the family is suggested to be included in the questionnaire. 
  4. Line 202, 'a total of 570 respondents but only accepted 423 responses', why other respondents were discarded? Not lived in Hanoi? Or any other criteria not met the study requirement? 
  5. Any comparison of Figure 1. (Frequency level of organic food consumption in households) in other cities in Vietnam or in Southeast Asia? 
  6. The resolution of Figure 3 should be improved. And the column color contrast of Figures 4 and 5 could be stronger. 
  7. Please define AMOS in line 12.
  8. A minor spell check is required for the manuscript.  eg. line 28.

Look forward to your responses and revisions!

Author Response

(The authors gave the same response as above.)

Reviewer 3 Report

Dear Authors,

Article entitled 'Organic food consumption among households in Hanoi: Importance of situational factors' represent an online survey-based report on sample respondents made on trends in organic food consumption by consumers in Hanoi. The article report valuable information as result and also the discussion and recommendation made by authors are valuable. However, there are certain issues in the presentation that must be taken care of which revising the manuscript for the improvement of the same. These include some basic errors like the citation of references and their formatting, improvement in the presentation of results, discussion, and conclusion as suggested/ highlighted in the attached Review pdf file.

Regards

Author Response

(The authors gave the same response as above.)

Reviewer 4 Report

This manuscript reports the analysis of factors influencing Vietnamese behavior in organic food purchase, addressing an issue of interest for the readers of Sustainability.

However, some points need to consider carefully before publishing.

First of all, please consider rephrasing the title of the manuscript (word „influence“ seems to be more appropriate than „importance“)

Add conclusion sentence(s) in the Abstract.

The Introduction is generally well written. However, the results of the previous, unpublished research should present more concisely.

A study aim(s) must be stated at the end of this section. 

Also, the Methodology section should be more concise, especially the part regarding statistical analysis.

The results in the Tables should present more clearly. Additionally, the data presented in various tables should not repeat in the text.

Tables should not insert in the Discussion.

The conclusion section also must be more concise with addressing future directions.

On the whole, I think that this manuscript needs some revision.

Author Response

(The authors gave the same response as above.)

Round 2

Reviewer 1 Report

Dear authors,

I acknowledge that you have made an important effort improving your manuscript. Nevertheless, I would like to draw your attention that my comments referred to the inclusion of a wider corpus of literature and a discussion of your results were not effectively solved. Below, I compiled some references that may be helpful in this.

Regarding the captions of tables and figures, I did not ask to remove the information you were presenting (cf. Figure 3), but to add some more. I find it very helpful to know how many people were represented and when were the data collected.

Best wishes,

References

Aertsens, J., Verbeke, W., Mondelaers, K., & Van Huylenbroeck, G. (2009). Personal determinants of organic food consumption: a review. British food journal,. 111(10), 1140-1167

Azzurra, A., Massimiliano, A., & Angela, M. (2019). Measuring sustainable food consumption: A case study on organic food. Sustainable Production and Consumption17, 95-107.

Monier, S., Hassan, D., Nichèle, V., & Simioni, M. (2009). Organic food consumption patterns. Journal of agricultural & food industrial organization7(2).

Napompech, K. (2019). Organic Food Purchase Motives of Southeast Asian Young Consumers. Asia-Pacific Social Science Review19(3), 270-279.

Nguyen, T. T. M., Phan, T. H., Nguyen, H. L., Dang, T. K. T., & Nguyen, N. D. (2019). Antecedents of purchase intention toward organic food in an asian emerging market: A study of urban vietnamese consumers. Sustainability11(17), 4773.

Author Response

Dear the reviewer,

Please receive our recently revised working paper. We would like to thank you a lot for your very useful and valuable comments which help us improve our working paper a lot.

Best regards,

Authors

Reviewer 3 Report

Dear authors,

There are still some errors in reference citation which need to be corrected before acceptance of the manuscript, if so. 

E.g.,

1. Reference in texts is to start in numerical order from [1] for Kim et al., 2012....and so on

2. Line 67-68:  Lee et al [Reference number, NOT THE YEAR]..., Likewise also the others.

3. Line 68: I guess it's Wang et al. (2020) not Wanga...please check and correct!

Line 83: Park and Lin [Reference No.]

Line 94: [31,44]

Line 116: delete dated

Please also see carefully the rest of the other citations in texts and also references cited therein. Please follow strictly the journal's guidelines in citing references.

Author Response

(The authors gave the same response as above.)

Reviewer 4 Report

The authors made the needed modifications to allow this paper to be acceptable. 

However, should cite references in the order in which they appear in the text. Also, according to the Sustainability citation format, use abbreviations for journal name rather than the full journal name.

Author Response

(The authors gave the same response as above.)
